# The Defined TLR3 Agonist, Nexavant, Exhibits Anti-Cancer Efficacy and Potentiates Anti-PD-1 Antibody Therapy by Enhancing Immune Cell Infiltration

**DOI:** 10.3390/cancers15245752

**Published:** 2023-12-08

**Authors:** Seung-Hwan Lee, Young-Ho Choi, Soon Myung Kang, Min-Gyu Lee, Arnaud Debin, Eric Perouzel, Seung-Beom Hong, Dong-Ho Kim

**Affiliations:** 1Research and Development Center, NA Vaccine Institute, Seoul 05854, Republic of Korea; shlee@navaccine.org (S.-H.L.); dudgh705@navaccine.org (Y.-H.C.); ksm91@navaccine.org (S.M.K.);; 2InvivoGen SAS, 5 Rue Jean Rodier, 31400 Toulouse, France; 3InvivoGen Ltd., Hong Kong Science and Technology Parks, Unit 307, 8W, Hong Kong, China

**Keywords:** Nexavant, TLR3 agonist, in situ vaccine, anti-tumor efficacy, immune checkpoint inhibitor, combination immunotherapy

## Abstract

**Simple Summary:**

Nexavant, a newly reported TLR3 agonist, has advantages over Poly(I:C) in terms of quality control and pre-clinical efficacy. Here, we further investigated the physicochemical properties, downstream signaling pathways, anti-cancer efficacy, and mechanism of action of Nexavant. The Nexavant was homogenous in solution and less sensitive to RNase A, and showed thermostability compared with Poly(I:C). Unlike Poly(I:C), which activates TLR3, RIG-I, and MDA5, Nexavant only activated TLR3 and RIG-I, not MDA5. The administration of Nexavant via either the intratumoral route or the intranasal route suppressed tumor growth in various cancer models. Combination therapy with anti-PD-1 antibodies resulted in better, synergistic tumor growth inhibition compared to the respective monotherapies. This study demonstrated that Nexavant could be more suitable for clinical use than Poly(I:C) and applied as an anti-cancer agent in the presence or absence of anti-PD-1 antibodies.

**Abstract:**

Nexavant was reported as an alternative to the TLR3 agonist of Poly(I:C) and its derivatives. The physicochemical properties, signaling pathways, anti-cancer effects, and mechanisms of Nexavant were investigated. The distinctive characteristics of Nexavant compared to that of Poly(I:C) were demonstrated by precise quantification, enhanced thermostability, and increased resistance to RNase A. Unlike Poly(I:C), which activates TLR3, RIG-I, and MDA5, Nexavant stimulates signaling through TLR3 and RIG-I but not through MDA5. Compared to Poly(I:C), an intratumoral Nexavant treatment led to a unique immune response, immune cell infiltration, and suppression of tumor growth in various animal cancer models. Nexavant therapy outperformed anti-PD-1 antibody treatment in all the tested models and showed a synergistic effect in combinational therapy, especially in well-defined cold tumor models. The effect was similar to that of nivolumab in a humanized mouse model. Intranasal instillation of Nexavant led to the recruitment of immune cells (NK, CD4+ T, and CD8+ T) to the lungs, suppressing lung metastasis and improving animal survival. Our study highlighted Nexavant’s defined nature for clinical use and unique signaling pathways and its potential as a standalone anti-cancer agent or in combination with anti-PD-1 antibodies.

## 1. Introduction

Since the approval of Yervoy, a CTLA-4-targeting antibody, for treating melanoma in 2011, immune checkpoint inhibitors (ICIs) such as anti-PD-1 and PD-L1 antibodies have been widely used to treat cancer [1,2]. However, apart from a small percentage of patients (approximately 10–35%) responding better to ICIs, many show minimal or no response. Additionally, some patients develop resistance to ICIs, resulting in cancer recurrence or accelerated cancer growth [3,4]. The main reason for the development of resistance to immuno-oncology drugs, including ICIs, is the increased secretion of immunosuppressive cytokines and angiogenic factors from immunosuppressive cells and the inhibition of immune cell infiltration into cancer cells (cold tumors, immuno-cold). Transforming a cold tumor, with inhibited immune cell infiltration, into a hot tumor, with more immune cells, can maximize the efficacy of immuno-oncology drugs and significantly improve treatment response rates [5].

To this end, research and clinical trials are underway to deliver pathogen-associated molecular patterns (PAMPs), such as Poly(I:C), to tumors [6,7,8,9]. PAMPs are recognized by pattern recognition receptors (PRRs), a protein found in the cell membrane or cytoplasm, inducing an innate immune response in the host. This allows a variety of immune cells to infiltrate the tumor and transform it into a hot tumor. In situ vaccines, injected directly into the tumor, can induce an immune response specific to an individual’s cancer by using a variety of antigens present in the cancer cells. Since these vaccines can use various antigens in cancer cells, they have the advantage of efficiently inducing immune responses to multiple cancer antigens and overcoming the heterogeneity of cancer cells.

Toll-like receptor 3 (TLR3) and retinoic-acid-induced gene-I (RIG-I)-like receptors (RLR) are well-known PRRs that recognize dsRNA and are highly expressed on innate immune cells such as DCs and macrophages [9,10,11]. In the immune system, these receptors serve as the first line of defense, allowing the host to recognize invading foreign pathogens with dsRNA. Their activation leads to the production of type I interferon (IFN), proinflammatory cytokines, and chemokines through the activation of interferon-stimulated gene (ISG) signaling, inducing the infiltration of various immune cells and a broad immune response [12,13].

A well-known representative dsRNA is polyinosinic: polycytidylic acid, abbreviated as Poly(I:C). It can strongly induce the secretion of IFN-β and inflammatory cytokines by activating TLR3, RIG-I, and MDA5 and can cause the infiltration of various immune cells [9,11,14,15]. In this regard, it is attracting attention as a vaccine adjuvant and an in situ vaccine. Poly(I:C) is synthesized poly-inosine and poly-cytidine conjugates with dsRNAs in the structure but has unordered complementarities, resulting in high heterogeneity [16,17]. The nature of Poly(I:C) significantly challenges stability, quality control, and monitoring in vivo PK. Various attempts were made to overcome these obstacles to improve the stability. Diverse formulations were attempted by adding additives to Poly(I:C) to produce Poly-ICLC, Poly-IC12U, PICKA, and BO-112 [9,16]. Since all of them use Poly(I:C) as the main component, the limitations of Poly(I:C) still exist. Some of the limitations were resolved in the length-defined candidates, such as RGC100, ARNAX, and TL-532, via chemical syntheses [16,18]. As shown in other studies [19], these molecules are not of the most optimized length for activating TLR3. Nevertheless, the potent efficacy of Poly(I:C) has encouraged researchers to conduct various clinical trials. In particular, Poly-ICLC (Hiltonol) is undergoing dozens of clinical trials as a vaccine adjuvant and an in situ vaccine for various patients of solid cancer [16,20].

We recently presented Nexavant as a well-defined TLR3 agonist [17]. Nexavant offers several advantages over Poly(I:C) in terms of quality control and clinical applicability. Nexavant comprises a 424 bp dsRNA core with five nt single-stranded overhangs at both 3′ ends, giving it a defined molecular weight of 275 kDa. Importantly, Nexavant has demonstrated stability under accelerated conditions (25 °C) for 6 months and a safety profile in several pre-clinical animal studies [17]. Unlike Poly(I:C), Nexavant of up to 99% purity can be produced and it allows scaled-up cost-effective synthesis. Moreover, it provides accurate pharmacokinetic and pharmacodynamic data, offering significant regulatory advantages [17].

An intramuscular administration of Nexavant to mouse systems notably increases the migration of dendritic cells (DCs), macrophages, and neutrophils to the draining lymph nodes [17]. Nexavant also triggers the upregulation of MHC-II, CD40, CD80, and CD86 in DCs, influencing DC maturation and activation. Nexavant holds promise as a vaccine adjuvant, as it efficiently stimulates dendritic cells, pivotal cells that impact vaccine efficacy. The activation of dendritic cells with Nexavant enhances their antigen-presenting capacity and cytokine secretion. It may ultimately increase cancer-antigen-specific T-cell responses and alterations in the cancer microenvironment, leading to enhanced anti-cancer effects.

In this study, we investigated the physicochemical properties, downstream signaling pathways, and anti-cancer efficacy of Nexavant compared with those of Poly(I:C). Simple methods can help precisely monitor the absolute amount and quality of Nexavant. In addition, the anti-cancer effects of Nexavant and Poly(I:C) via intratumoral or intranasal administration were compared in various animal models. Considering the enhanced anti-cancer effect, signaling pathway, and possible molecules involved, Nexavant was proposed as an alternative tool for anti-cancer therapy of cold tumors. Furthermore, the efficacy of anti-PD-1 antibody therapy was synergistically enhanced when combined with intratumoral administration of Nexavant, especially in the cold tumor models. Nexavant treatment led to immune cell infiltration and suppressed tumor growth via a unique innate immune response.

## 2. Materials and Methods

### 2.1. Animals

Female C57BL/6 or BALB/c mice at 6–8 weeks of age were purchased from Samtako Bio Korea (Osan-si, Gyeonggi-do, Republic of Korea), and hPD-1-knockin C57BL/6 or BALB/c mice were purchased from GH Bio (Daejeon, Republic of Korea). The mice were maintained at the NA Vaccine Institute (NAVI) animal facility (Seoul, Republic of Korea); fed a sterile, commercial mouse diet; and provided with water ad libitum. The experimental protocols used in this study were reviewed and approved by the NAVI’s Ethics Committee and Institutional Animal Care and Use Committee (IACUC).

### 2.2. Cell Lines

The B16F10 murine melanoma cell line was purchased from Korean Cell Line Bank (Seoul, Korea), and the CT26 murine colon cancer cell line, the LL/2 murine lung cancer cell line, and the 4T-1 murine breast cancer cell line were purchased from ATCC. B16F10, LL/2, and EMT6 cells were maintained in DMEM (Welgene, Gyeongsan-si, Gyeongsangbuk-do, Republic of Korea); and CT26 and 4T-1 cells were maintained in RPMI (Welgene). All DMEM and RPMI media were supplemented with 10% fetal bovine serum (FBS) (Welgene) and 1% penicillin/streptomycin (Gibco) and incubated at 37 °C in 5% CO_2_.

HEK-Dual RNA 4KO hTLR3, HEK-Dual RNA 4KO hRIG-I, and HEK-Dual RNA 4KO MDA5 cell lines were developed by InvivoGen (San Diego, CA, USA). They are derived from the human embryonic kidney 293 (HEK293)-Dual cell line harboring the stable integration of two inducible reporter genes for secreted embryonic alkaline phosphatase (SEAP) and Lucia luciferase. As a result, these cells allow the interferon regulatory factor (IRF) pathway by assessing the activity of a secreted luciferase.

Briefly, HEK-Dual RNA 4KO cells were obtained using standard genome editing procedures to suppress the gene expression of TLR3, RIG-I, MDA5, and PKR. The complete absence of IRF responses in these cells upon stimulation with dsRNAs was functionally validated. HEK-Dual RNA 4KO cells were engineered to express only one dsRNA sensor stably, using TLR3-, RIG-I-, or MDA5-encoding plasmids. The expression of SEAP, Lucia, and PRRs was maintained by growing the cells in media containing blasticidin (ant-bl-05, InvivoGen), Zeocin (ant-zn-05, InvivoGen), and Puromycin (ant-pr-1, InvivoGen). Cell lines were routinely checked for mycoplasma contamination using PlasmoTest (rep-pt-1, InvivoGen).

### 2.3. Reagents and Antibodies

Nexavant was produced using in vitro transcription following the procedures described in the previous report [17]. Poly(I:C) HMW (tlrl-pic for in vitro assay and vac-pic for in vivo assay), Poly(I:C) LMW (Cat. #: tlrl-picw), the lipid-based transfection reagent LyoVec (Cat. #: lyec-12), the luciferase detection reagent QUANTI-Luc 4 Lucia/Gaussia (Cat. #: rep-qlc4lg1), and the SEAP detection reagent QUANTI-Blue 4 (Cat. #: rep-qbs) were purchased from InvivoGen. Cell-staining antibodies for the flow cytometry analysis, including PE/Cy7-conjugated anti-mouse CD11b mAb (Cat. #: 101215), APC-conjugated anti-mouse CD11c mAb (Cat. #: 117310), FITC-conjugated anti-mouse CD19 mAb (Cat. #: 115506), FITC-conjugated anti-mouse CD335 mAb (Cat. #: 137606), PE/Cy5-conjugated anti-mouse CD3 mAb (Cat. #: 100309), PE/Cy5-conjugated anti-mouse CD45 mAb (Cat. #: 103110), PerCP/Cyanine5.5-conjugated anti-mouse CD45 mAb (Cat. #: 103132), PE-conjugated anti-mouse CD80 mAb (Cat. #: 104708), PE/Cy5-conjugated anti-mouse CD86 mAb (Cat. #: 105016), APC/Cyanine7-conjugated anti-mouse CD8 mAb (Cat. #: 100714), FITC-conjugated anti-mouse CD8 mAb (Cat. #: 100705), PE-conjugated anti-mouse F4/80 mAb (Cat. #: 123110), AF700-conjugated anti-mouse I-A/I-E mAb (Cat. #: 107621), PerCP/Cy5.5-conjugated anti-mouse Ly6G mAb (Cat. #: 127615), and 7-AAD Viability Staining Solution (Cat. #: 420404) were purchased from BioLegend (San Diego, CA, USA).

### 2.4. Treatment with Nexavant and Poly(I:C)

When the mice were administered Nexavant and Poly(I:C), two different routes were used, depending on the experiment. For intratumoral administration, 50 μL of Nexavant and Poly(I:C) was slowly injected into the center of the tumor using an Ultra-Fine II syringe (BD, Franklin Lakes, NJ, USA). For nasal delivery to the lungs, the mice were anesthetized by the inhalation of isoflurane, and 30 μL of Nexavant and Poly(I:C) was slowly instilled dropwise through the nostrils.

### 2.5. ISG Signaling Induction

In a 96-well plate, 5 × 10^4^ HEK-Dual RNA 4KO hTLR3, HEK-Dual RNA 4KO hRIG-I, or HEK-Dual RNA 4KO MDA5 cells were cultured overnight with increasing concentrations of Poly(I:C) HMW (Cat. #: tlrl-pic, InvivoGen), Poly(I:C) LMW (Cat. #: tlrl-picw, InvivoGen), or Nexavant, either naked (10 µg/mL to 10 ng/mL) or pre-complexed (1 µg/mL to 1 ng/mL) with the LyoVec transfection reagent according to the manufacturer’s instructions. Recombinant human IFN-β (1000 U/mL to 1 U/mL, ThermoFisher, Waltham, MA, USA) was used as the positive control for IRF activation. According to the manufacturer’s instructions, the ISG response was measured by monitoring the Lucia luciferase activity in the culture supernatants using QUANTI-Luc™ 4 Lucia/Gaussia.

### 2.6. Tumor Models

C57BL/6 mice were subcutaneously inoculated with 5 × 10^5^ cells of B16F10 or LL/2 in their flanks, and BALB/c mice were inoculated with CT26 cells. BALB/c mice were inoculated with 4T-1, EO771, or EMT6 cells in their mammary fat pads. Nexavant or Poly(I:C) HMW (vac-pic, InvivoGen) was injected into the tumors of the mice, and an anti-mouse PD-1 antibody (Cat. #: BE0146, BioXCell, Lebanon, NH, USA) was injected intraperitoneally at the same time. Tumor size was measured with a caliper three to four times a week, and tumor volume was calculated using the following equation: (width × length × height) × 3.14/6. The percentage of tumor growth inhibition (TGI) was calculated using the following equation: {(tumor volume of control group)—(tumor volume of sample group)/(tumor volume of control group)} × 100. The mice were euthanized when they exhibited signs of poor health or when the tumor volume exceeded about 2000 mm^3^. To establish the lung metastasis model, 5 × 10^5^ cells of B16F10 were injected through the tail vein of C57BL/6 mice. As per the experimental schedule, the lung metastasis mice were sacrificed and the lungs were harvested. The gross photograph of a lung with tumor burden was taken with a digital camera (Sony α5100), and the tumor nodules of the lungs were counted.

### 2.7. Analysis of Immune Cells from Inguinal Lymph Nodes and Tumors

The inguinal lymph nodes were collected at the indicated times from the mice injected with Nexavant and crushed into a 70 μm strainer. Next, the cells of the lymph nodes were dissociated with collagenase type IV (Worthington Biochemical Co., Lakewood, NJ, USA) for 20 min at 37 °C, washed with PBS containing 1% FBS and 0.1% sodium azide (Sigma Aldrich, St. Louis, MO, USA), and subjected to flow cytometry. The tumors were collected from the B16F10-tumor-bearing mice on day 13 and crushed into a 70 μm strainer. Next, the tumor cells were dissociated with collagenase types IV and I (Worthington Biochemical Co.) for 20 min at 37 °C and washed with PBS containing 1% FBS and 0.1% sodium azide (Sigma Aldrich). The cells were stained with antibodies as described above. Unstained samples were used as negative controls. Stained cells were analyzed using a NovoCyte flow cytometer (Agilent Technologies, Santa Clara, CA, USA).

### 2.8. Enzyme-Linked Immunosorbent Assay (ELISA)

Blood samples were collected from the mice treated with Nexavant or the mice bearing tumors. Serum was separated from the blood samples after centrifugation at 13,000 rpm for 10 min. The levels of IFN-β, IL-6, and IL-12 in the serum were quantified with ELISA kits according to the manufacturer’s instructions (Invitrogen, Waltham, MA, USA).

### 2.9. Statistical Analysis

Data were analyzed using Prism V6 (GraphPad, San Diego, CA, USA) and represented as mean values ± the standard error of the mean (SEM). One-way ANOVA and two-way ANOVA with Dunnett’s test were performed to compare more than two groups.

## 3. Results

### 3.1. Physicochemical Advantage of Nexavant as a Homogeneous TLR3 Agonist

We compared the physicochemical properties of Nexavant and Poly(I:C). First, the feasibility of quantitation using simple UV absorbance spectrometry was tested. Poly(I:C) was notably evident with relatively low accuracy (coefficient of variation 29.48%). The peak absorbance was observed between 250 and 270 nm. The absorption spectrum of Poly(I:C) exhibited two maxima, one at λ = 248 nm and one at λ = 269 nm (Figure 1A). These characteristics are probably due to the low uniformity within the solution and the heterogeneous structure of Poly(I:C), which consists of a mixture of single- and double-stranded RNA with varying chain lengths. Nexavant measurements were highly accurate (coefficient of variation 1.56%), and their absorbance profiles resembled those of typical nucleic acids, showing a peak at 260 nm. Unlike Poly(I:C), the quality and quantity of Nexavant can be accurately monitored at any time using simple UV spectrometry and agarose gel analyses.

We observed that Nexavant displayed lower sensitivity to RNase A than Poly(I:C) (Figure 1B). While Poly(I:C) underwent degradation at low doses of RNase A concentrations of 0.005 U/mL, Nexavant remained resistant to RNase A even at higher concentrations. The susceptibility of Poly(I:C) to RNase A could be attributed to the presence of single-stranded RNA generated during the manufacturing process due to incomplete hybridization between poly-I and poly-C chains (Figure 1B). In contrast, the reduced sensitivity of Nexavant to RNase A can be attributed to its dsRNA structure with complete complementary hybridization. When each molecule was stored at 42 °C for thermostability assay, Poly(I:C) exhibited structural changes or material alterations within a day. Nexavant remained stable after 1 week (Figure 1C) and after 6 months. These findings highlight that Nexavant is a well-defined TLR3 agonist in terms of its physical and structural properties. It can be consistently produced, exhibits low sensitivity to RNase A, and demonstrates long-term stability at non-freezing temperatures.

### 3.2. Similar but Distinct TLR3 Pathway Activation of Nexavant

MDA5 detects long-duplex RNAs (>500 bp) in the genome of double-stranded RNA (dsRNA) viruses or dsRNA replication intermediates of viruses [21,22,23]. In contrast, RIG-I detects the five triphosphate group (5′ppp) and the blunt ends of short dsRNAs (<500 bp) or single-stranded RNA (ssRNA) hairpins [24,25]. To determine the cellular action mechanisms of Nexavant, we applied HEK-293 dual-reporter model cell lines for interferon-stimulating signaling, which express either TLR3, RIG-I, or MDA5 genes ectopically on the basis of genetic deficiency of four different endogenous dsRNA sensor genes (i.e., TLR3, RIG-I, MDA5, and PKR). Both the lipoplexed and naked forms of NVT and Poly(I:C) (both HMW and LMW forms) induced TLR3- and RIG-I-dependent signaling in a dose-dependent manner (Figure 2A,B, respectively). The strength of the signaling was higher when the treatment was with the lipoplexed forms compared with when the treatment was with the naked counterparts. Contrary to Poly(I:C) (both HMW and LMW forms), which strongly induced MDA5-dependent signaling when lipoplexed, even the lipoplexed NVT did not induce MDA5-dependent signaling (Figure 2C). This is an expected result since NVT would not be binding to MDA5 due to its relatively short dsRNA length (<500 bp). In the case of RIG-I, shorter dsRNAs induced stronger activation in a dsRNA-dependent manner (Figure 2B). For MDA5, Poly(I:C) HMW and LMW induced strong activation, while NVT did not induce MDA5 activation even with LyoVec (Figure 2C). Consistent with other studies showing the MDA5 activation and dsRNA length dependency [21], Nexavant is a novel TLR3 agonist that activates only RIG-I, not MDA5. Overall, Nexavant can be defined as a distinct TLR3 agonist with moderate potency and a unique activator of the molecular cascade system compared with Poly(I:C).

### 3.3. Intratumoral Delivery of Nexavant Induces the Recruitment of Various Immune Cells into the Tumor

The immune cell profile in the inguinal lymph nodes and cytokine levels in the blood were monitored over 48 h after a single subcutaneous injection of Nexavant in the flank of a healthy mouse (Figure 3A). Most of the analyzed immune cells, including CD4+ and CD8+ T cells, B cells, neutrophils, NK cells, and macrophages, were recruited to the lymph nodes, although the time to peak and the kinetics were specific to the cell type. The total number of dendritic cells in the lymph node seemed to show a gradual increase over time; however, statistical significance was not attained due to high sample-to-sample variations. The expression of DC activation markers, such as CD80 and CD86, exhibited an increase after 4 h, reached its peak at 24 h, and then declined. Nexavant may exert a stronger effect on activation compared with the recruitment of DC. Detectable serum levels of IFN-β and proinflammatory cytokines, such as IL-6 and IL-12, typical products of ISG signaling, were induced at 4 h after injection (Figure 3B). While both IFN-β and IL-6 declined to undetectable levels in 12 h, the IL-12 remained detectable for a longer duration and declined to the baseline level in 48 h.

To examine the anti-cancer efficacy of Nexavant compared to that of Poly(I:C), each was administered to mouse tumors in the melanoma model (Figure 3C). While both suppressed tumor growth, Nexavant suppressed tumor growth better than Poly(I:C) at the same dose (100 μg/animal; tumor growth inhibition (TGI)% for Nexavant 83.18% and TGI% for Poly(I:C) 41.21%). To understand why Nexavant exhibits better anti-cancer efficacy, we examined the levels of cytokines induced in the serum and immune cell infiltration into tumors after both treatments. The levels of IL-6 and IFN-β in the blood were higher in the group treated with Nexavant than in the group treated with Poly(I:C) (Figure 3D). Immune cells such as macrophages, NK, DC, and CD8+ T cells were significantly increased in the tumors in both groups (Figure 3E). However, no statistical differences were found between the two groups.

The results demonstrate that both Nexavant and Poly(I:C) in the tumor tissue induce infiltration of diverse immune cells. However, unlike in the case of Poly(I:C), when Nexavant is administered, there is enhanced secretion of cytokines, such as IFN-β and IL-6, which may induce a unique tumor microenvironment for improved anti-cancer effect.

### 3.4. Intratumoral Delivery of Nexavant Shows Anti-Tumor Effects in Various Tumor Models

To investigate whether Nexavant shows anti-cancer effects beyond mouse melanoma, tumor growth inhibition with Nexavant was tested in various mouse cancer models (Figure 4A). Nexavant suppressed tumor growth by 61.5, 76.15, and 58.04% in LL/2 lung cancer, CT26 colon cancer, and 4T-1 breast cancer models, respectively. This indicates that Nexavant has broad anti-cancer effects on a variety of cancers. The dose–response and abscopal effect of Nexavant was examined in a mouse melanoma model (B16F10) (Figure 4B). Intratumoral administration of Nexavant at doses as low as 25 μg and as high as 150 μg resulted in significant tumor growth inhibition in both injected (72.93–96.49%) and non-injected distant tumors (TGI% 62.09–74.73%). However, the dose–response was not evident in this dose range.

To investigate whether Nexavant is effective in the late stage of the tumor, we began administering Nexavant 22 days after the inoculation of tumor cells (mean tumor volume 750 mm^3^) and maintained the treatment schedule twice weekly. While 100 μg/animal was ineffective, 500 μg/animal effectively controlled tumor growth (Figure 4C). In this late-stage tumor-bearing model, tumor growth was suppressed when a higher dose was administered. This shows that the dose of Nexavant for intratumoral application should be varied depending on the volume and stage of the tumor. The presented data demonstrate that Nexavant has anti-cancer activity against a wide range of cancers and the tumor status, such as tumor size and stage, can have an influence on what dose is effective.

### 3.5. The Anti-Tumor Effect of Anti-PD-1 Antibody Therapy Can Be Improved by Delivering Nexavant Intratumorally

We used mouse melanoma and TNBC models, known as cold tumors [26,27], to determine whether the intratumoral administration of Nexavant combined with anti-PD-1 antibodies improves the anti-cancer effect. In the mouse melanoma model, the anti-PD-1 antibody and Nexavant monotherapy groups showed a TGI effect of 35.88 and 78.33%, respectively. The combination therapy group showed a TGI effect of 96.19% (Figure 5A). Each treated animal harboring a tumor was rechallenged with mouse melanoma cells in the flank of the opposite side on day 24 and monitored for tumor growth. No tumor growth from the rechallenge was observed in the groups pre-treated with the therapy combining anti-PD-1 antibodies and Nexavant.

Intratumoral immune cell analysis demonstrated that compared with monotherapies with anti-PD-1 antibodies and Nexavant, combining the two significantly increased NK cells, CD4, and CD8 T cells (Figure 5B). The additive/synergistic effect was demonstrated not only in B16F10 melanoma but also in K1735 melanoma (Figure 5C). In the K1735 melanoma model, Nexavant and anti-PD-1 antibodies inhibited tumor growth by 60.42% and 31.27%, respectively, and the combination significantly improved the inhibition effect to 89.46%. We also tested the effectiveness of the combination therapy in triple-negative breast cancer (TNBC), a well-known model of cold tumors (Figure 5D,E). Compared to the monotherapies, the combination significantly improved the tumor inhibition effect in both models. Although the degree of the anti-cancer effect of Nexavant was different depending on the type of the tumor, it was confirmed that the combination therapy significantly improved the anti-cancer effect of anti-PD-1 antibodies. The synergistic effect of dual therapy was evident in the melanoma and TNBC models, which are representative examples of cold tumors.

### 3.6. Intratumoral Administration of Nexavant Also Has a Synergistic Effect with Nivolumab, a Human Anti-PD-1 Antibody

A clinically relevant, fully human anti-PD-1 antibody (i.e., nivolumab) was employed to investigate the anti-cancer efficacy of combination therapy. We used a genetically engineered mouse expressing a chimeric PD-1 with the extracellular domain of human PD-1 and the transmembrane and cytoplasmic domains of mouse PD-1. Similar to the previous study involving the use of an anti-PD-1 antibody of murine origin, the combination of Nexavant and nivolumab was found to be more effective than the respective monotherapies in both the B16F10 melanoma and the EMT6 TNBC models (Figure 6A,B, respectively). While nivolumab alone showed low tumor growth inhibition (49.85%), similar to the anti-PD-1 antibody of murine origin, the combination therapy dramatically improved tumor growth inhibition and survival. The combinational effect in the EMT6 TNBC model was impressive, with an 83% complete response. This further supports the theory that therapy with a combination of Nexavant and anti-PD-1 antibodies has improved anti-cancer efficacy, especially in cold tumor models.

### 3.7. Intranasal Administration of Nexavant Has a Potent Anti-Tumor Effect on the Lung Metastasis Model

Poly(I:C) was shown to have an anti-cancer effect on the lung cancer model through lung delivery [28]. To broaden the application of Nexavant, we tested whether Nexavant also has an anti-cancer effect when delivered to the lung metastasis model. We first examined how the lung’s immune cell profile changes when Nexavant is administered to the lung (Figure 7A). Flow cytometric analysis of the bronchoalveolar lavage fluid (BALF) cells harvested 24 h after Nexavant administration demonstrated that NK and CD8+ T cells were significantly increased. No significant changes were found for macrophages, neutrophils, and CD4+ T cells. In the B16F10 lung metastasis model, Nexavant or Poly(I:C) was instilled on days 8 and 14, and metastatic lung nodules were examined on day 18. Compared to the lungs of the Poly(I:C)-treated mice, the lungs of the Nexavant-treated mice showed a reduced number and size of tumor nodules (Figure 7B), and the Nexavant group showed improved survival compared to the Poly(I:C) group (Figure 7C).

To investigate whether the stage of tumor development affects the anti-cancer effects of Nexavant, it was instilled every other day starting on day 2 (early-treatment group), day 8 (intermediate-treatment group), or day 14 (late-treatment group) after cancer cell injection through the tail vein and the survival rates of the animals were assessed (Figure 7D). Nexavant prolonged the survival of the animals when they were treated earlier (days 2 and 8) but did not when the animals were treated late (day 14). On day 15, the number of tumor nodules in the lungs of the mice in the early-treatment group, which responded best to Nexavant, was reduced about 7-fold compared with the controls (Figure 7E). Given the synergistic effects of intratumorally administered Nexavant and anti-PD-1 antibodies, we tested whether intranasally administered Nexavant also enhances the efficacy of anti-PD-1 antibody treatment. While Nexavant monotherapy showed strong tumor growth suppression compared with anti-PD-1 antibody monotherapy, the combination therapy did not exhibit better tumor growth suppression compared with the monotherapies (Figure 7F). These results suggest that Nexavant alone could be sufficiently effective in some contexts and may not require to be administered in combination with anti-PD-1 antibodies.

## 4. Discussion

Poly(I:C) and its derivatives have been used as TLR3 agonists for more than a decade for diverse studies and clinical developments [9,16,18]. Despite a considerable number of studies, the clinical products based on them are minimal. Notwithstanding their clinical efficacy, it is hard to control product homogeneity, and thus their quantification could be challenging. Because Poly(I:C) is formed by incomplete annealing of poly-inosine and poly-cytidine chains, it is highly heterogeneous in length and structure, resulting in a heterogeneous macromolecule. Due to this heterogeneity, there is sizable batch-to-batch variability in average length, composition, and in vivo stability, which may result in variability in efficacy.

Nexavant, a newly reported TLR3 agonist, is a dsRNA with a defined length and structure with a high uniformity [17]. Nexavant is a perfectly matched dsRNA with a specified length of 424 bp, which is expected to be more stable in vitro and in vivo than Poly(I:C). It was shown that Nexavant is more resistant to RNase A digestion and shows thermostability at 42 °C (Figure 1B,C). The homogeneous nature of Nexavant allows for better production quality control, which is expected to result in more consistent efficacy. Most of all, the amount and quality of Nexavant can be monitored using a simple method of UV absorbance or agarose gel electrophoresis. Since Nexavant can be stored at 42 °C for up to 6 months [17], storing the reagent is convenient and logistically energy and cost-efficient.

Shorter forms of Poly(I:C) (<500 bp) produced via the RNase digestion of Poly(I:C) HMW are able to activate RIG-I, which requires short dsRNA (<500 bp), but not MDA5, which requires long dsRNA (>500 bp) [21,22,23]. Since Nexavant is shorter than 500 bp, it only activated RIG-I, not MDA5 (Figure 2). Accordingly, Nexavant should be classified as a distinct agonist activating TLR3 and RIG-I but not MDA5. As PKR can be activated by short dsRNA (>30 bp), NVT is a likely activator of PKR [29]. However, due to the technical difficulties of generating PKR-dependent model cell lines, NVT’s activity related to PKR was not tested in this study. We previously reported that when Nexavant was administered into mice, the mRNA levels of TLR3, RIG-I and MDA5 were increased, suggesting that Nexavant may act as a dsRNA ligand similarly to Poly(I:C) [17]. It is speculated that the induction of MDA5 signaling observed in vivo is due to a positive feedback effect resulting from the activation of TLR3 and RIG-I signaling. In contrast to Poly(I:C) HMW, Nexavant induced relatively low TLR3 and RIG-I ISG signaling activation. Interestingly, it showed more significant tumor growth inhibition than Poly(I:C) in mouse tumor models (Figure 3C). This is likely due to differences in the in vivo stability and mode of action of Nexavant and Poly(I:C).

Intratumoral administration of Nexavant induced the recruitment of various immune cells to the administration site (Figure 3E). Nexavant may stimulate antigen-presenting cells (APCs) to capture diverse cancer antigens and migrate to draining lymph nodes to present cancer antigen information to T cells. Cancer-antigen-specific T cells circulate throughout the body via the blood or lymphatic vessels. They infiltrate cancerous tissue, where the cancer antigen is present, and engage with cancer cells bearing the antigen. Eventually, the antigen-specific T cells become activated and eliminate the cancer cells. Due to their systemic circulation, these T cells have the potential to target and eradicate not only the primary cancer site but also metastatic cancers, a phenomenon known as the abscopal effect. When Nexavant is injected into a tumor, the immune cell infiltration pattern resembles the pattern of injecting Poly(I:C) (Figure 3E and citation [30]). Based on the difference in efficacy and the levels of IFN-β and IL-6, the potent anti-cancer effect of Nexavant may need to be further investigated (Figure 3D). Also, it is interesting that Nexavant can be applied to the terminal stage of the cancer animal model with a dose increment (Figure 4C).

The most promising immune checkpoint inhibitors of the current clinical practice are anti-PD-1 antibodies [31,32]. PD-1 antagonists work by disrupting the T cell immune evasion mechanisms used by cancer cells, thereby enhancing the anti-cancer effect. However, it was reported that the effectiveness of immune checkpoint inhibitors, such as anti-PD-1 antibodies, is notably diminished in “cold” tumors, characterized by a scarcity of immune cells within the tumor microenvironment [33,34]. When anti-PD-1 antibodies and Nexavant are co-administered intratumorally, diverse immune cells are infiltrated into the tumor (Figure 5B) and shown to have a synergistic anti-cancer effect (Figure 5). Nexavant with anti-PD-1 antibodies can synergistically enhance the recruitment of immune cells to cold tumors.

In this study, we observed that the anti-cancer efficacy of anti-PD-1 antibodies varied depending on the cancer animal model (partially effective in EO771 and B16F10 models; less effective in K1735 and EMT6 models). However, the anti-cancer efficacy of Nexavant was more consistent across the test animal models. When administered intratumorally, Nexavant showed a consistent anti-cancer effect in most subcutaneous tumor models, and combination therapies involving Nexavant and anti-PD-1 antibodies were additive/synergistic in most models. This result suggests that Nexavant can be applicable as a monotherapy or a combination therapy, overcoming the limited clinical efficacy of anti-PD-1 antibodies. To expand the idea to a more clinically relevant application, a human anti-PD-1 antibody, nivolumab, was tested in transgenic mice expressing a chimeric PD-1 with the extracellular domain of human PD-1. The partial anti-cancer effect of Nexavant and nivolumab was synergistic in combinational therapy. Interestingly, the dual therapy was as effective as 83% of the complete response in a TNBC cancer model.

The intranasal administration of Nexavant demonstrated a robust anti-cancer activity against lung metastases. The increase in the number of immune cells in the lungs following Nexavant treatment is believed to underlie the anti-cancer effect with recruited groups of immune cells, which can directly induce changes in the tumor microenvironment (Figure 7A). This assay also indicates the better potency of Nexavant than Poly(I:C). The study demonstrated that for optimal effectiveness, Nexavant should be administered during the early stages to the mid-stages of cancer progression. Unlike an intratumoral application, the intranasal combination therapy with anti-PD-1 antibodies did not show improvement. There could be several reasons for this, such as the anatomical features of the lung and the distribution of anti-PD-1 antibodies to the lung.

## 5. Conclusions

In this study, we demonstrated that, compared with Poly(I:C), Nexavant is a distinct TLR3 agonist that possesses better physicochemical properties, i.e., a feasible quantitation and qualification, thermostability, and more resistance to RNase A for enhanced charters for mass manufacturing and clinical use. Unlike the Poly(I:C) HMW form, which targets all three dsRNA sensors, i.e., TLR3, RIG-I, and MDA5 in both the naked and cationic-lipid-complexed forms, Nexavant targets only TLR3 in the naked form and both TLR3 and RIG-I, but not MDA5, in the cationic-lipid-complexed form. In the case of in situ intratumoral application, Nexavant itself proves to be a potent anti-cancer reagent in diverse cancer animal models by recruiting immune cells into tumor tissue. Its synergistic effect with anti-PD-1 antibodies was demonstrated in various cancer models. The combination of nivolumab with a chimeric mouse with a human PD-1 domain also has a synergistic effect and suppresses tumor growth by recruiting diverse immune cells into tumors, especially in a cold tumor model. On intranasal application, Nexavant shows a potent anti-cancer effect on the metastatic lung cancer model. However, contrary to the synergistic effect of the combination of Nexavant and anti-PD-1 antibodies visible in the intratumoral model, this synergistic effect was not observed in the lung metastasis model.

## Figures and Tables

**Figure 1 cancers-15-05752-f001:**
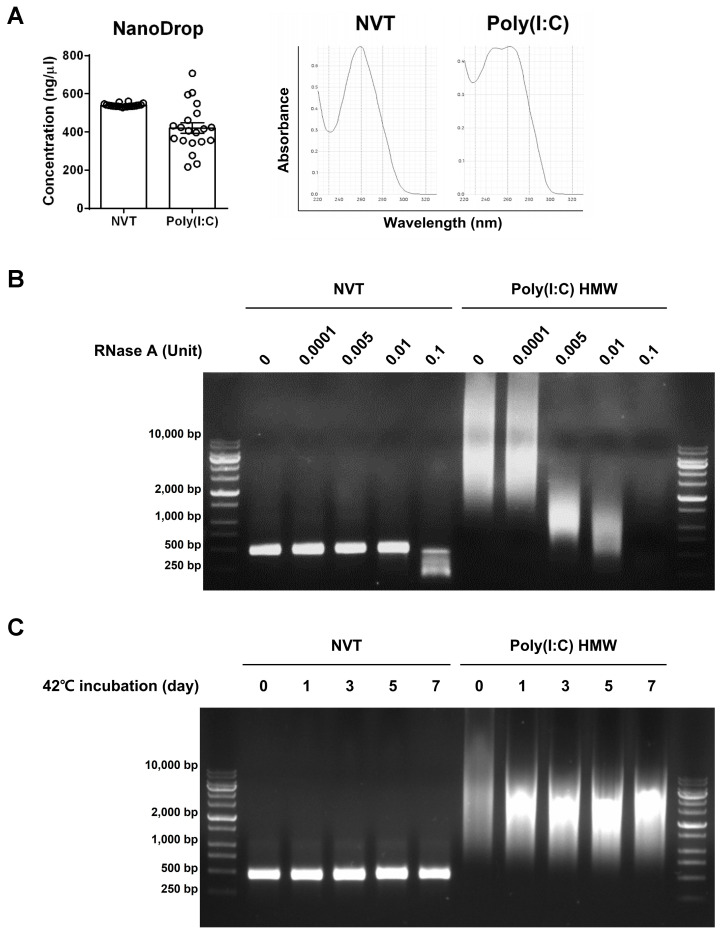
Physicochemical characteristics of Nexavant (NVT) and Poly(I:C) as demonstrated using UV spectrometric analysis, stability to RNase A, and thermostability. (**A**) Accuracy of quantification using UV spectrometry (A260 nm) and profiles of UV absorbance spectra (NanoDrop, *n* = 20). (**B**) Susceptibility to various concentrations of RNase A at 37 °C for 10 min. (**C**) Susceptibility to different incubation times at 42 °C. The uncropped blots are shown in Appendix A.

**Figure 2 cancers-15-05752-f002:**
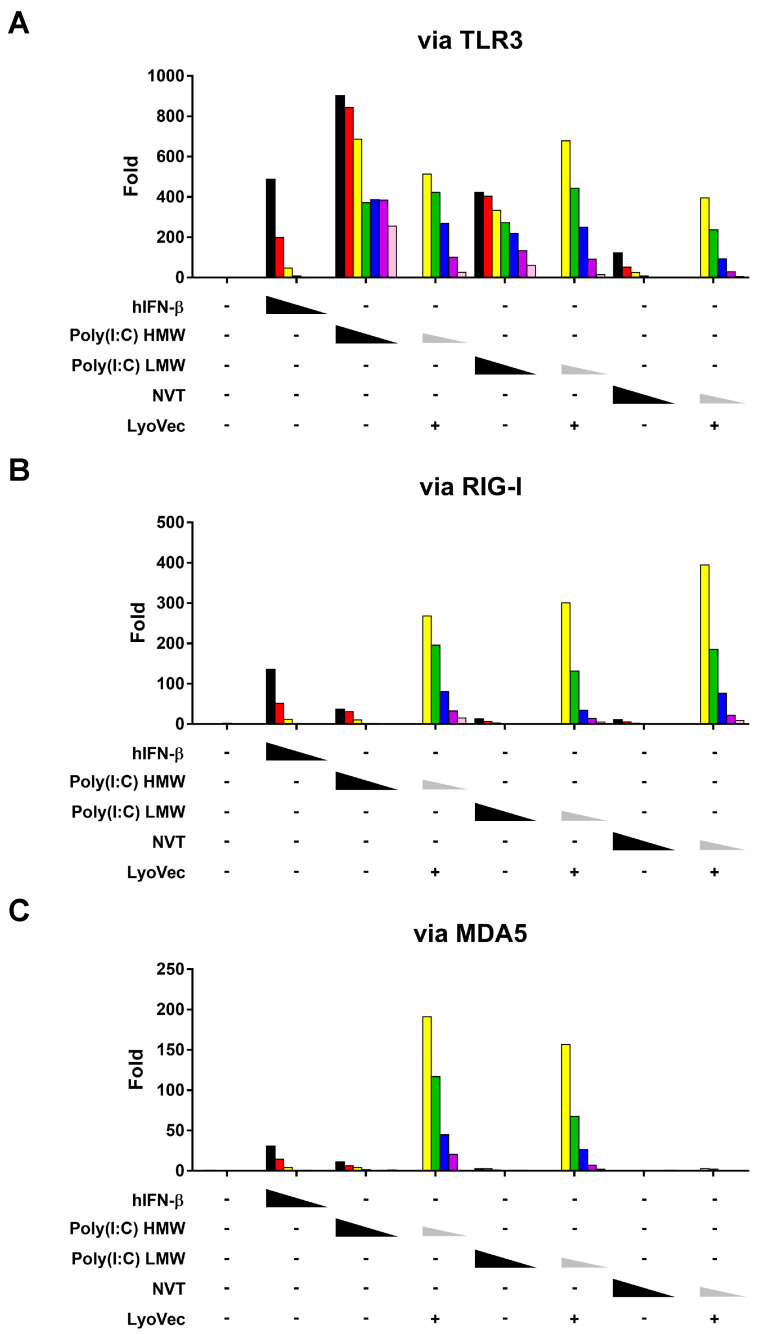
The naked Nexavant and the cationic-lipid-complexed Nexavant regulate the downstream signaling pathways. The three model cell lines expressing TLR3, RIG-I, or MDA5 genes were stimulated with naked or cationic-lipid-complexed Nexavant to examine its mechanism of action. Responsive reporter systems measured dose-dependent induction of (**A**) TLR3, (**B**) RIG-I, and (**C**) MDA5 signaling. As controls, hIFN-β, hIL-1β, Poly(I:C) HMW, and Poly(I:C) LMW were used. The fold inductions for RLU and absorbance compared with the non-treated groups were calculated. hIFN-β, Poly(I:C) HMW, Poly(I:C) LMW, and Nexavant were treated as follows: hIFN-β: 1000, 300, 100, 30, 10, 3, and 1 unit/mL; Poly(I:C) HMW, Poly(I:C) LMW, or Nexavant: 10,000, 3000, 1000, 300, 100, 30, and 10 ng/mL; Poly(I:C) HMW, Poly(I:C) LMW, or Nexavant (with LyoVec): 1000, 300, 100, 30, 10, 3, and 1 ng/mL. Data are representative of two independent experiments.

**Figure 3 cancers-15-05752-f003:**
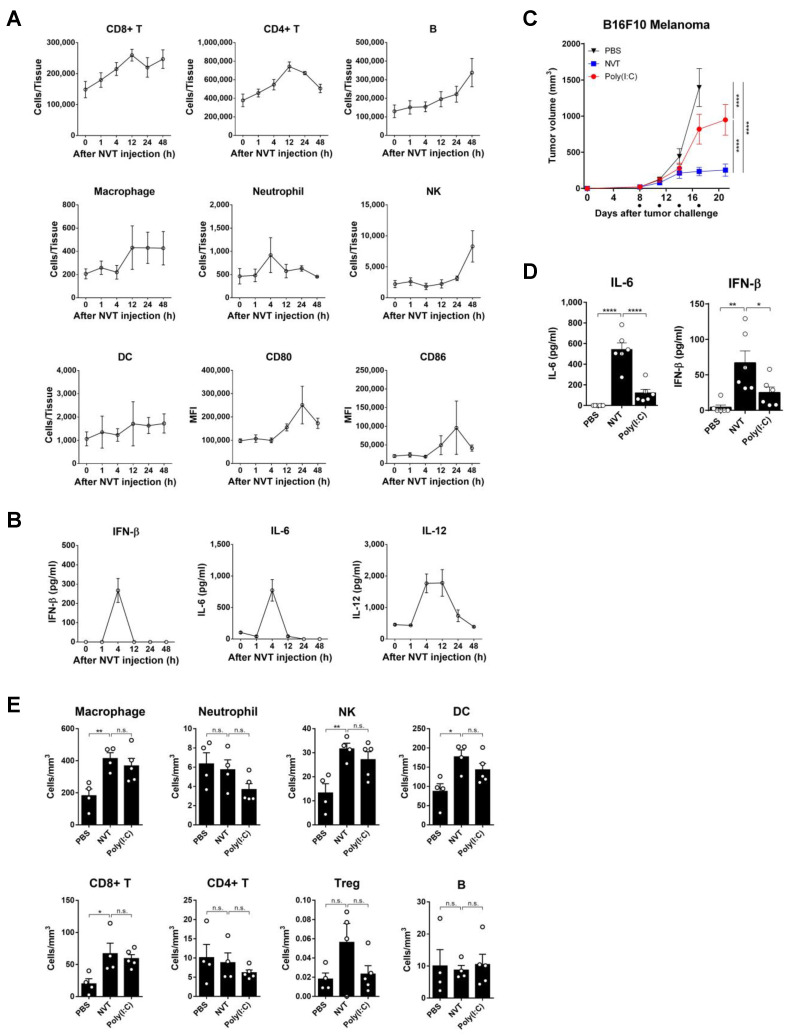
Nexavant recruits diverse immune cells to the draining lymph nodes near the injection sites. C57BL/6 mice (*n* = 4/group) were injected subcutaneously with 25 μg of Nexavant. At 1, 4, 12, 24, and 48 h post-injection, the mice were monitored for (**A**) immune cell profile and DC activation in the inguinal lymph nodes and (**B**) IFN-β, IL-6, and IL-12 cytokine levels in the blood. C57BL/6 mice (*n* = 5/group) inoculated subcutaneously with 5 × 10^5^ B16F10 cells were treated with 100 μg of Nexavant or Poly(I:C) on days 8, 11, 14, and 17 post-inoculation, and (**C**) tumor growth was measured. (**D**) Blood levels of IL-6 and IFN-β cytokines were measured 4 h after the first injection of Nexavant or Poly(I:C) (*n* = 6/group). (**E**) The immune cell profile of mouse tumors was analyzed on day 15 (*n* = 4, 4, 5). * *p* < 0.05, ** *p* < 0.01, and **** *p* < 0.0001; n.s., not significant; two-way ANOVA with Dunnett’s test (**C**) and one-way ANOVA with Dunnett’s test (**D**,**E**).

**Figure 4 cancers-15-05752-f004:**
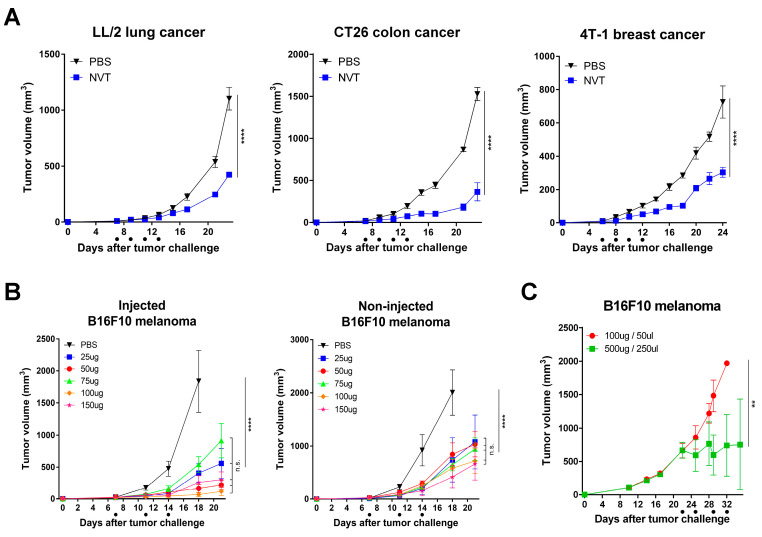
Intratumoral administration of Nexavant suppressed tumor growth in various syngeneic models. (**A**) C57BL/6 mice (*n* = 4/group) inoculated subcutaneously with 5 × 10^5^ LL/2, CT26, or 4T-1 cells were treated with Nexavant (25 μg) (LL/2 and CT26 models on days 7, 9, 11, and 13 post-inoculation; the 4T-1 model on days 6, 8, 10, and 12 post-inoculation). (**B**) C57BL/6 mice (*n* = 5/group) were inoculated subcutaneously in both flanks with 5 × 10^5^ B16F10 cells. Nexavant (25, 50, 75, 100, and 150 ugs) was injected into the right flank tumor 7-, 11-, and 14-days post-inoculation. (**C**) C57BL/6 mice (*n* = 4/group) inoculated subcutaneously with 5 × 10^5^ B16F10 cells were treated with Nexavant (100 μg and 500 μg) on days 22, 25, 29, and 32 post-inoculation. ** *p* < 0.01, and **** *p* < 0.0001; n.s., not significant; two-way ANOVA with Dunnett’s test (**A**–**C**).

**Figure 5 cancers-15-05752-f005:**
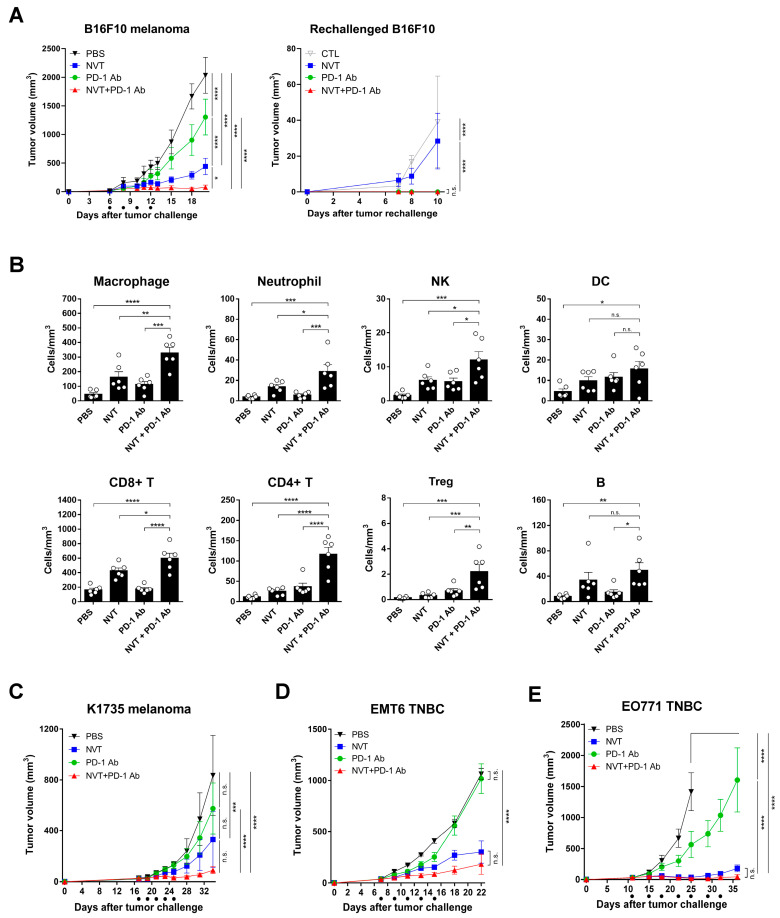
Combination therapy with anti-PD-1 antibodies and Nexavant shows an improved anti-tumor effect. (**A**) C57BL/6 mice (*n* = 5/group) were inoculated subcutaneously in their right flanks with 5 × 10^5^ B16F10 tumor cells and treated with 25 μg of Nexavant (intratumoral) and 100 μg of anti-PD-1 antibodies (intraperitoneal) on days 6, 8, 10 and 12 post-inoculation. The mice were inoculated again, this time in their left flanks, with 5 × 10^5^ B16F10 tumor cells on day 24 after the first challenge (corresponding to day 0 of the rechallenge). Since all individuals in the PBS group died before day 24, naïve mice were used as a control group (CTL) for rechallenge tumors. The volume of injected and non-injected tumors was measured at the indicated time points. (**B**) The immune cell profiles of the B16F10 tumors were assessed on day 25 using flow cytometry (*n* = 6/group). (**C**) C57BL/6 mice (*n* = 5/group) were subcutaneously inoculated in their right flanks with 5 × 10^5^ K1735 tumor cells and immunized with Nexavant and anti-PD-1 antibodies on days 17, 19, 21, 23, and 25 post-inoculation. (**D**) BALB/c mice (*n* = 5/group) were inoculated in their mammary fat pads with 5 × 10^5^ EMT6 tumor cells and then immunized with Nexavant and anti-PD-1 antibodies on days 7, 9, 11, 13, and 15 post-inoculation. (**E**) C57BL/6 mice (*n* = 5/group) were inoculated in their mammary fat pads with 5 × 10^5^ EO771 tumor cells and then immunized with Nexavant and anti-PD-1 antibodies on days 11, 15, 18, 22, 25, 29, and 32 post-inoculation. Tumor volumes were measured at indicated time points. * *p* < 0.05, ** *p* < 0.01, *** *p* < 0.001, and **** *p* < 0.0001; n.s., not significant; two-way ANOVA with Dunnett’s test (**A**,**C**–**E**) and one-way ANOVA with Dunnett’s test (**B**).

**Figure 6 cancers-15-05752-f006:**
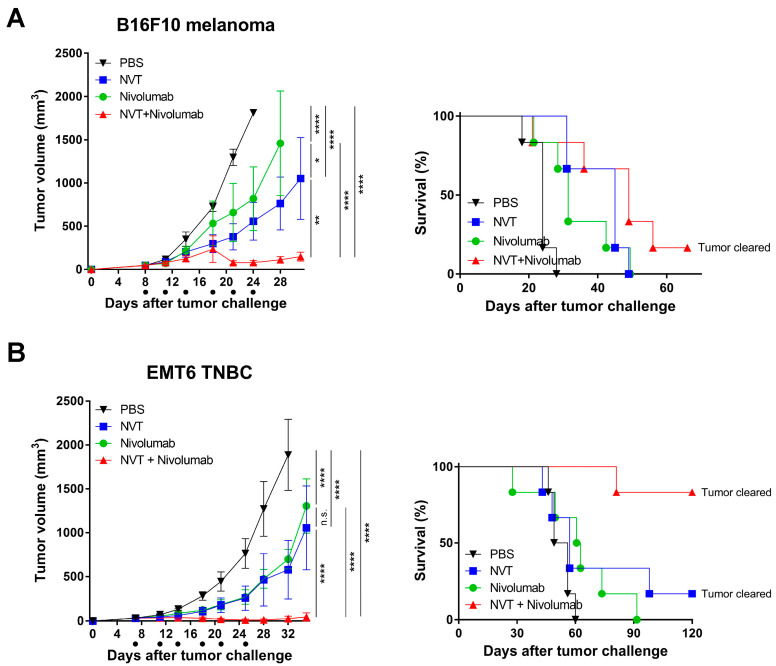
Improved anti-cancer efficacy of the combined treatment with Nexavant and nivolumab (a human anti-PD-1 antibody) was tested in the transgenic mice expressing a chimeric PD-1 with the extracellular domain of human PD-1 and the transmembrane and cytoplasmic domains of mouse PD-1 compared with the respective monotherapies. C57BL/6 and BALB/c mice with human PD-1 were inoculated with 5 × 10^5^ B16F10 melanoma cells and EMT6 TNBC cells, respectively (day 0). (**A**,**B**) Tumor growth inhibition and improved survival in the group administered the combination therapy compared with the groups administered monotherapies in (**A**) B16F10 melanoma and (**B**) EMT6 breast cancer models (*n* = 6/group). B16F10-bearing mice were treated with Nexavant and anti-PD-1 antibodies on days 8, 11, 14, 18, 21, and 24 post-inoculation, and EMT6-bearing mice were treated on days 7, 11, 14, 18, 21, and 25 post-inoculation. Tumor volumes and survivals were measured at indicated time points. * *p* < 0.05, ** *p* < 0.01, and **** *p* < 0.0001; n.s., not significant; two-way ANOVA with Dunnett’s test (**A**,**B**).

**Figure 7 cancers-15-05752-f007:**
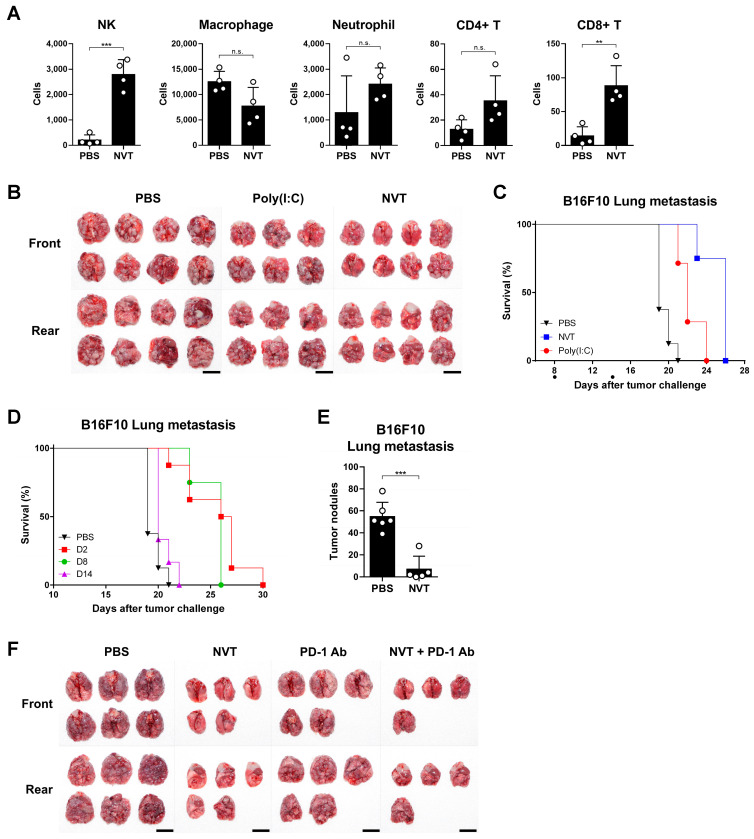
Intranasal administration of Nexavant has a potent antitumor effect but there is no additive effect when it is combined with anti-PD-1 antibodies. (**A**) In all, 25 μg of Nexavant was slowly instilled through the nostrils (*n* = 4/group). After 24 h, bronchoalveolar lavage fluid (BALF) was collected and analyzed using flow cytometry. (**B**,**C**) C57BL/6 mice were injected with 5 × 10^5^ B16F10 cells via the tail vein. Next, 25 μg of Nexavant or Poly(I:C) was slowly instilled through the nostrils every two days starting on day 8 or 14. Tumor-bearing lungs ((**B**); *n* = 8, 6, 8/group) were harvested on day 18, and overall survival ((**C**); *n* = 8, 8, 7) was monitored. (**D**,**E**) C57BL/6 mice were injected with 5 × 10^5^ B16F10 cells via the tail vein. Next, 25 ug of Nexavant was slowly instilled through the nostrils every two days starting on days 2, 8, or 14 (*n* = 8, 8, 8, 6). (**E**) On day 15, the lungs were harvested from B16F10 lung metastases, and the tumor nodules were counted (*n* = 6, 5). (**F**) C57BL/6 mice (*n* = 6, 5, 5, 4) were injected with 5 × 10^5^ B16F10 cells via the tail vein. Next, 25 μg of Nexavant was instilled via the nostrils, and 200 μg of anti-PD-1 antibodies was injected intraperitoneally on days 2, 4, 6, 8, and 10. On day 18, B16F10 lung metastasis mice were sacrificed and the lungs were harvested. The gross photograph of a lung with tumor burden was taken from a certain distance. Scale bar = 1 cm; ** *p* < 0.01, and *** *p* < 0.001; n.s., not significant; one-way ANOVA with Dunnett’s test (**A**,**E**). The uncropped blots are shown in Appendix A.

## Data Availability

The data presented in this study are available in this article and Appendix A.

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
