# Peer review of "The Defined TLR3 Agonist, Nexavant, Exhibits Anti-Cancer Efficacy and Potentiates Anti-PD-1 Antibody Therapy by Enhancing Immune Cell Infiltration"

_cancers, 2023, doi:10.3390/cancers15245752_

Round 1

Reviewer 1 Report

Comments and Suggestions for Authors

The study entitled “The defined TLR3 agonist, Nexavant, exhibits anti-cancer efficacy and potentiates anti-PD-1 antibody therapy by enhancing immune cell infiltration” by Lee et al showed Nexavant as a distinct activator of TLR3 and RIG-I signaling, and combination therapy with anti-PD-1 antibody exhibited synergistic tumor growth inhibition. The study is well designed with proper controls.

It has been previously shown that Nexavant exhibits TLR3, MDA-5, and RIG-I signaling (PMID: 36761735), but in the current study authors have reported the activation of only TLR3 and RIG-I signaling by Nexavant. It is not clear why MDA-5 is not induced by Nexavant. Does Nexavant exhibit differential effect on different models?

Comments on the Quality of English Language

There are significant changes needed to the style and grammar of the manuscript.

Author Response

Described in L509-513.

‘We previously reported that when mice were administered Nexavant, it increased mRNA levels of TLR3, RIG-I, and MDA5, suggesting that Nexavant may act as a dsRNA ligand similar to Poly(I:C)17. It is speculated that the induction of MDA5 signaling observed in vivo is due to a positive feedback effect on the activation of TLR3 and RIG-I signaling.’

There are significant changes needed to the style and grammar of the manuscript.

Get the English editing service.

Reviewer 2 Report

Comments and Suggestions for Authors

In this preclinical study the authors present data to show that the synthetic dsRNA TLR3 agonist Nexavant exerts anti-tumour activity in several murine models alone or in combination with anti-PD1 Tx that exceeds that seen with Poly I:C. They provide a convincing argument of the superiority of Nexavant over Poly I:C in terms of in vivo stability, homogeneity and ability to induce anti-tumour effects alone or in combination with ICT.

I will say at the outset that this study merits publication but only after a number of improvements have been made to the manuscript as follows:-

MAJOR POINTS

In my view there needs to be a more robust and extensive statistical analysis undertaken on some of the results as presented. Examples here are Figure 1C where comparison should be made of the statistical significance of differences between each treatment group. The same can be said of Figs 4A/4B (note standard deviations are large in some cases), Figs 5C/5D & 5E, Fig 6A/6B. Additionally the Kaplan-Meier survival curves in Figs 6A/6B and 7C should be statistically analysed using log-rank analysis to determine the significance of differences between the treatment groups. All of this data should be made clear in both the results narrative and in figure legends together with a statement of the number of animals used in each treatment group. 

Additional to this, where statistical analysis has been undertaken (e.g. Fig 3D/3E and 5B) a key to the star system used to show significant differences between groups should be incorporated into the figure legends.

Under the section 3.2 there is no reference made to Figure 2 in the results narrative. In this section the authors claim that Nexavant (lines 260-261) "turned out to be a decent TLR3 and RIG-1 activator regardless of the presence of the lipid". This statement is somewhat nebulous and furthermore is not really supported by the data presented in Figs 2A and 2B where it seems to me that there is only a weak activation. This needs rewording in a manner that is appropriate to the data.

The authors should emphasise more in both the results and discussion sections that the effects they see with both Poly I:C and Nexavant are DOSE-DEPENDENT as demonstrated in their presented data. They seem to fail to get this across strongly enough in my view. 

MINOR POINTS

In general the standard of scientific English is good throughout the paper. However there are a number of incorrect word usages as follows:

line 163 treatment "WITH" not of

line 460 replace qualification with QUALITATION

line 461 replace feasible with ACHIEVABLE

line 545 replace Unlikely with UNLIKE

This preclinical study has a direct clinical relevance and as such should be published in Cancers after the suggested improvements have been satisfactorily made.

Comments on the Quality of English Language

None

Author Response

In my view there needs to be a more robust and extensive statistical analysis undertaken on some of the results as presented. Examples here are Figure 1C where comparison should be made of the statistical significance of differences between each treatment group. The same can be said of Figs 4A/4B (note standard deviations are large in some cases), Figs 5C/5D & 5E, Fig 6A/6B. Additionally the Kaplan-Meier survival curves in Figs 6A/6B and 7C should be statistically analysed using log-rank analysis to determine the significance of differences between the treatment groups. All of this data should be made clear in both the results narrative and in figure legends together with a statement of the number of animals used in each treatment group. 

Diversify the statistic method and modify the legends and marked in red.

Additional to this, where statistical analysis has been undertaken (e.g. Fig 3D/3E and 5B) a key to the star system used to show significant differences between groups should be incorporated into the figure legends.

The legend was modified and marked in red.

Under the section 3.2 there is no reference made to Figure 2 in the results narrative. In this section the authors claim that Nexavant (lines 260-261) "turned out to be a decent TLR3 and RIG-1 activator regardless of the presence of the lipid". This statement is somewhat nebulous and furthermore is not really supported by the data presented in Figs 2A and 2B where it seems to me that there is only a weak activation. This needs rewording in a manner that is appropriate to the data.

An additional description was made in the Fig2 result section (L254-275) as follows.

MDA5 detects long-duplex RNAs (> 500 bp) in the genome of double-stranded RNA (dsRNA) viruses or dsRNA replication intermediates of viruses21-23. In contrast, RIG-I de-tects the five triphosphate group (5′ppp) and the blunt ends of short dsRNAs (<500 bp) or single-stranded RNA (ssRNA) hairpins24, 25. To determine the cellular action mechanisms of Nexavant, we applied HEK-293 dual-reporter model cell lines for interferon stimulating signaling, which express either TLR3, RIG-I, or MDA5 genes ectopically on the basis of genetic deficiency of four different endogenous dsRNA sensor genes (i.e., TLR3, RIG-I, MDA5, and PKR). Both the lipoplexed and naked forms of NVT and Poly(I:C) (both HMW and LMW forms) induced TLR3- and RIG-I-dependent signaling in a dose-dependent manner (Figure 2A and 2B, respectively). The strength of the signaling was higher when the treatment was with the lipoplexed forms compared with when the treatment was with the naked counterparts. Contrary to Poly(I:C) (both HMW and LMW forms), which strongly induced MDA5-dependent signaling when lipoplexed, even the lipoplexed NVT did not induce MDA5-dependent signaling (Figure 2C). It is an expected result since NVT would not be binding to MDA5 due to its relatively short dsRNA length (<500 bp). In the case of RIG-I, shorter dsRNAs induced stronger activation in a dsRNA-dependent manner (Figure 2B). For MDA5, Poly(I:C) HMW and LMW induced strong activation, while NVT did not induce MDA5 activation even with LyoVec (Figure 2C). Consistent with other studies showing the MDA5 activation and dsRNA length dependency21, Nexavant is a novel TLR3 agonist that activates only RIG-I, not MDA5. Overall, Nexavant can be defined as a distinct TLR3 agonist with moderate potency and a unique activator of the molecular cascade system compared with Poly(I:C).’

The authors should emphasise more in both the results and discussion sections that the effects they see with both Poly I:C and Nexavant are DOSE-DEPENDENT as demonstrated in their presented data. They seem to fail to get this across strongly enough in my view. 

There was no significant dose-response. It was modified in L337-340 as follows.

‘Intratumoral administration of Nexavant at doses as low as 25 ug and as high as 150 ug resulted in significant tumor growth inhibition in both injected (72.93-96.49%) and non-injected distant tumors (TGI% 62.09-74.73%). However, the dose response was not evident in this dose range.’

MINOR POINTS

In general the standard of scientific English is good throughout the paper. However there are a number of incorrect word usages as follows:

Had the English editing service.

Reviewer 3 Report

Comments and Suggestions for Authors

The manuscript is described briefly, however, some typing mistakes and some sentences are confusing and require to be clarified. For example, L503, L547, L551, L93-100 without citation in this paragraph.

other questions are as follows:

1. Section 3.2, PKR result and Figure 2 are missed.

2. How is the TGI% determined?

3. L284, the number of DC is remained but the markers of DC  are increasing over time. How would the authors discus this observation?

4. Fig 4C, the mean tumor volume was 750mm3, but in other figures, tumor volume was about 2000mm3 on around 20-22 days.

5. What is “non-injected B16F10 melanoma” in Fig. 4B?

6. What is CTL curve in Fig.5A?

Comments on the Quality of English Language

some typing mistakes.

Author Response

  1. Section 3.2, PKR result and Figure 2 are missed.

An additional explanation was added in Fig2 (L254-275).

‘MDA5 detects long-duplex RNAs (> 500 bp) in the genome of double-stranded RNA (dsRNA) viruses or dsRNA replication intermediates of viruses21-23. In contrast, RIG-I de-tects the five triphosphate group (5′ppp) and the blunt ends of short dsRNAs (<500 bp) or single-stranded RNA (ssRNA) hairpins24, 25. To determine the cellular action mechanisms of Nexavant, we applied HEK-293 dual-reporter model cell lines for interferon stimulating signaling, which express either TLR3, RIG-I, or MDA5 genes ectopically on the basis of genetic deficiency of four different endogenous dsRNA sensor genes (i.e., TLR3, RIG-I, MDA5, and PKR). Both the lipoplexed and naked forms of NVT and Poly(I:C) (both HMW and LMW forms) induced TLR3- and RIG-I-dependent signaling in a dose-dependent manner (Figure 2A and 2B, respectively). The strength of the signaling was higher when the treatment was with the lipoplexed forms compared with when the treatment was with the naked counterparts. Contrary to Poly(I:C) (both HMW and LMW forms), which strongly induced MDA5-dependent signaling when lipoplexed, even the lipoplexed NVT did not induce MDA5-dependent signaling (Figure 2C). It is an expected result since NVT would not be binding to MDA5 due to its relatively short dsRNA length (<500 bp). In the case of RIG-I, shorter dsRNAs induced stronger activation in a dsRNA-dependent manner (Figure 2B). For MDA5, Poly(I:C) HMW and LMW induced strong activation, while NVT did not induce MDA5 activation even with LyoVec (Figure 2C). Consistent with other studies showing the MDA5 activation and dsRNA length dependency21, Nexavant is a novel TLR3 agonist that activates only RIG-I, not MDA5. Overall, Nexavant can be defined as a distinct TLR3 agonist with moderate potency and a unique activator of the molecular cascade system compared with Poly(I:C).’

Additional discussion was made for PKR in L502-509.

‘Shorter forms of Poly(I:C) (<500 bp) produced via the RNase digestion of Poly(I:C) HMW are able to activate RIG-I, which requires short dsRNA (<500 bp), but not MDA5, which requires long dsRNA (>500 bp)21-23. Since Nexavant is shorter than 500 bp, it only activated RIG-I, not MDA5 (Figure 2). Accordingly, Nexavant should be classified as a dis-tinct agonist activating TLR3 and RIG-I but not MDA5. As PKR can be activated by short dsRNA (>30 bp), NVT is a likely activator of PKR29. However, due to the technical difficul-ties of generating PKR-dependent model cell lines, NVT’s activity related to PKR was not tested in this study.’

  1. How is the TGI% determined?

The additional description was made in Materials and Methods/2.6 (L188-191).

‘The percentage of tumor growth inhibition (TGI) was calculated using the following equation: {(tumor volume of control group) - (tumor volume of sample group) / (tumor volume of control group)} × 100.’

  1. L284, the number of DC is remained but the markers of DC are increasing over time. How would the authors discus this observation?

Modified in (L297-298).

‘Nexavant may exert a stronger effect on activation compared with the recruitment of DC.’

  1. Fig 4C, the mean tumor volume was 750mm3, but in other figures, tumor volume was about 2000mm3 on around 20-22 days.

Based on the suggestion of IACUC, the subpopulation of mice with less tumor size was used in this assay based on ethical issues. The initial tumor size of these groups was less than 5% of the variation.

  1. What is “non-injected B16F10 melanoma” in Fig. 4B?

The word was changed in L335-337 to describe the tumor that Nexavant did not directly inject to show an abscopal effect.

  1. What is CTL curve in Fig.5A?

Additional description in Legend (L394-395).

‘Since all individuals in the PBS group died before day 24, naïve mice were used as a control group (CTL) for rechallenge tumors.’

Comments on the Quality of English Language some typing mistakes.

English editing service

Round 2

Reviewer 1 Report

Comments and Suggestions for Authors

The manuscript has been sufficiently improved for the publication in Cancers.

Reviewer 3 Report

Comments and Suggestions for Authors

The manuscript has been revised. No further comment.